# Computerized Assessment of Occlusion and Muscle Activity during Use of a Multilayer Clear Retainer: A Preliminary Study

**DOI:** 10.3390/s21020541

**Published:** 2021-01-13

**Authors:** Kyoung Yeon Kim, Jin-Young Choi, Song Hee Oh, Hyung-Wook Moon, Seong-Hun Kim, Hyo-Won Ahn, Kyung A Kim, Gerald Nelson

**Affiliations:** 1Department of Orthodontics, Graduate School, Kyung Hee University, Seoul 02447, Korea; iamnalda@empas.com (K.Y.K.); joyful.ortho@gmail.com (J.-Y.C.); wooky29@hanmail.net (H.-W.M.); hyowon@khu.ac.kr (H.-W.A.); k2aortho@khu.ac.kr (K.A.K.); 2Department of Oral and Maxillofacial Radiology, Graduate School, Kyung Hee University, Seoul 02447, Korea; ohbbang50@gmail.com; 3Division of Orthodontics, Department of Orofacial Science, University of California San Francisco, San Francisco, CA 94143, USA; gdnelson41@gmail.com

**Keywords:** occlusion, orthodontic retainer, T-scan, electromyography, muscle activity

## Abstract

The aim of this preliminary study was to evaluate the short-term changes of occlusal contacts and muscle activity after orthodontic treatment during the use of a multi-layer clear retainer. Evaluation was done with the T-scan and BioEMG systems. A total of 18 subjects were included, who were evaluated at three time intervals—T0 at debonding, T1 at one month after retainer delivery, and T2 at four months after retainer delivery. The T-scan and electromyography (EMG) data were recorded simultaneously. The T-scan system recorded the occlusion time, disclusion time and force distribution. The EMG waves were quantified by calculating the asymmetry index and activity index. The time variables changed but not significantly. Occlusal force decreased in the anterior dentition and increased in the posterior dentition during T0–T2. There was no clear evidence of a relationship between unbalanced occlusal forces and muscle activity. In most subjects, the temporalis anterior muscle was more dominant than the masseter muscle. From this preliminary computerized study, there were no significant changes in the state of the occlusion or muscle activity during the short-term retention period.

## 1. Introduction

After orthodontic treatment, the retention phase is intended to maintain the corrected occlusion and function. Without retention, one expects relapse, or unfavorable change from the final occlusion [1]. A favorable change after orthodontic treatment also occurs, called settling. Teeth naturally will erupt toward each other to find stable points of contact, improving intercuspation and masticatory function. With settling, the number of occlusal contacts increases [2]. Dental occlusion and occlusal forces have been suggested as one of the factors for stability since the development of contemporary orthodontics. Edward H. Angle, known as the father of modern orthodontics, stressed the importance of occlusal factors for the post-treatment stability [3].

After achieving the desired occlusion through orthodontic treatment, several types of removable retainers, such as the Hawley retainer, circumferential retainer, or a vacuum formed retainer (VFR) are prescribed and applied to patients. Comparative studies have been reported of the effects of various removable retainers on retention. Demir et al. compared the VFR with the Hawley retainer [4] and showed that the VFR was more effective to retain the mandibular anterior teeth during the retention period. Rowland et al. [5] suggested a similar result: that the VFRs had more effective influence on the retention of mandibular incisors. While the VFR showed good post-treatment retention, occlusal settling is interfered with. Moreover, the clear aligner, similar to the VFR, causes posterior open bite because of the posterior interocclusal plastic layers [6]. In addition to the prevention of occlusal settling, the VFR has poor wear resistance and durability of the occlusal surfaces, and the materials of this retainer have dimensional instability.

To overcome these limitations, a new type of clear retainer was developed called the Oral-treaper (OTP) (Figure 1) [7,8]. It is composed of multiple layers of polymers. The outer layer was made of modified polyethylene terephthalate glycol (PETG), which has good fatigue resistance and dimensional stability [9]. The middle layer is thermoplastic polyurethane (TPU) to increase the elasticity and relieve the forces delivered to the appliance [10,11]. The inner part is a reinforced resin core covering the incisal and lingual sides of the anterior teeth and the occlusal surfaces of the posterior teeth. The inner layer does not extend to the second molars to ensure settling of those teeth. Kim et al. [8] evaluated the settling pattern of patients during the use of the OTPs and showed an improvement in posterior occlusal settling. Most important in any study of settling patterns is the study methodology. Kim’s study measured the occlusion according to the depth of the occlusal contacts in digital models, which were based on study models from alginate impressions.

To evaluate the occlusion, measurements of occlusal contacts, force, and timing are required. Traditional occlusal analysis methods such as articulating paper, shim stock, waxes and silicone impressions are referred to as qualitative methods. They are incomplete due to their static nature, subjective interpretation, and lack of reliability and reproducibility [12]. A quantitative occlusal analysis can provide much better data. For example, the T-scan system (Tekscan Inc., South Boston, MA, USA) is a diagnostic device that records bite force dynamics, including relative occlusal force, location, and timing. When patients bite the pressure-measuring sensors, the occlusal changes are displayed directly on the computer screen and recorded. The force distribution ratios for each tooth are displayed on a 2D and 3D graph (Figure 2). Electromyography (EMG) is a diagnostic method using electrical potentials that is the most objective and reliable technique for evaluating muscle function and efficiency (Figure 3). With this device, the activation signal of muscles can be evaluated and recorded. It catches the electrical signals associated with the contraction. Kerstein [13] suggested the use of the T-scan and EMG systems in performing occlusal adjustment procedures. The simultaneous recording of both systems allows analyzing and correlating specific occlusal moments to specific muscle changes.

There have been a few studies that evaluate the relationship of occlusal contacts and masticatory muscles in post-orthodontic subjects, however they have not considered the retention protocol [14,15]. To the best of the authors’ knowledge, there are no studies investigating both occlusal dynamics and EMG muscle activity during post-orthodontic period follow-up. The aim of the present study was to evaluate the innocuous effects of the OTP retainer to occlusal changes in the retention period after orthodontic treatment, by evaluating the changes in the occlusion and masticatory muscle activity with a T-scan and EMG. The null hypothesis of this study was that the use of the OTP retainer does not affect the occlusion as expressed through a T-Scan and EMG during the retention period.

## 2. Materials and Methods

The selected patients finished their orthodontic treatment at the Department of Orthodontics in Kyung Hee University Dental Hospital. The study design was reviewed and approved by the Institutional Review Board of Kyung Hee University Dental Hospital (IRB No.: KHD IRB 1612-7).

### 2.1. Subjects

The inclusion criteria for the subjects were patients: (1) who had full fixed orthodontic treatment, including upper and lower braces (0.022-in Quicklear brackets with Tweemac prescription, Forestadent, Bernard Förster GmbH, Pforzheim, Germany); (2) who had fully erupted second molars; (3) who had no more than four missing teeth excluding third molars; and (4) who had no edentulous spaces within the arches. The exclusion criteria were patients: (1) who had missing teeth gaps or dental implants; (2) who had experienced jaw surgery; (3) who had anterior or posterior cross-bite or open-bite; (4) who had signs or symptoms of temporomandibular disorder (TMD); or (5) who had a CO (centric occlusion)–CR (centric relation) discrepancy. A total of 18 patients participated in this study. At debonding (T0), one month after retainer delivery (T1), and four months after retainer delivery (T2), the occlusion and muscle activity were inspected. The demographic data of participants is described in Table 1.

The sample size was calculated with the G*Power version 3.1.9.7 software (University of Düsseldorf, Düsseldorf, Germany) [16,17]. A priori power analysis suggested a minimum sample size of 16 subjects in F tests with three time measurements at an α = 0.05, power of 0.8, and calculated effect size of 0.33 under partial η^2^ = 0.1.

### 2.2. Retention Protocols

At the completion of orthodontic treatment, lingual retainers were bonded on maxillary and mandibular anterior teeth. Two weeks later, they received an Oral-Treaper (OTP) retainer for the maxillary dentition and a tongue elevator for the mandibular dentition (Figure 4). As described above, the OTP is a clear-overlay type retainer which is composed of three layers (Figure 1) [7,8]. The outer layer is polyethylene terephthalate glycol (PETG), the middle layer is thermoplastic polyurethane (TPU), and an inner layer is a reinforced resin core. The tongue elevator is an appliance that elevates the patient’s tongue to a correct tongue posture [18]. It is composed of a lower anterior labial bow, occlusal rests on all molars, and acrylic resin ledges below the tongue. Patients were instructed to wear the retainers during sleeping.

### 2.3. Occlusal and EMG Analysis

At each time point, subjects underwent occlusal and muscle activity analysis with the T-scan Novus ver.9 system and BioEMG III system (BioResearch Inc., Milwaukee, WI, USA) in the multifactor malocclusion service (MMS) clinic of Kyung Hee University Dental Healthcare Center (Figure 2 and Figure 3). All recordings were made with the patient in an upright position in the unit chair. After the skin was cleaned with 70% isopropyl rubbing alcohol, EMG electrodes were placed onto the TA and MM bilaterally. The T-scan sensor was also placed in the subject’s mouth. The T-scan and EMG recordings were simultaneously recorded (Figure 5).

The T-scan recorded the occlusion time (OT) and force distribution in a maximal intercuspal position (MICP) and the disclusion time (DT) in protrusive and left/right excursive movements. OT is the time taken from initial tooth contact to MICP, and DT is the time from when anterior or lateral guidance is formed to the point when molar contact is lost. The EMG waves were quantified by calculating the asymmetry index and activity index for each subject at maximum occlusal force of MICP position. All mandibular movements were measured a total of 3 times each and the average of 3 values was calculated.

### 2.4. Statistical Analysis

The original asymmetry index describes the asymmetry between the right and left sides of paired muscles [19]. We modified the asymmetry index so that the positive value of the asymmetry index indicated which side of the mouth had a higher occlusal force and a higher muscle activity value. The negative value of asymmetry index indicates the side of lower occlusal force and muscle activity.
$$\mathrm{Asymmetry}\;\mathrm{Index}=\frac{{\mathrm{MM}}_{\mathrm{High}}+{\mathrm{TA}}_{\mathrm{High}}-{\mathrm{MM}}_{\mathrm{Low}}-{\mathrm{TA}}_{\mathrm{Low}}}{{\mathrm{MM}}_{\mathrm{High}}+{\mathrm{TA}}_{\mathrm{High}}+{\mathrm{MM}}_{\mathrm{Low}}+{\mathrm{TA}}_{\mathrm{Low}}}\times 100\;\left(\%\right)$$

Activity index describes the relative contribution of the MM and TA [19].
$$\mathrm{Activity}\;\mathrm{Index}=\frac{{\mathrm{MM}}_{\mathrm{Right}}+{\mathrm{MM}}_{\mathrm{Left}}-{\mathrm{TA}}_{\mathrm{Right}}-{\mathrm{TA}}_{\mathrm{Left}}}{{\mathrm{MM}}_{\mathrm{Right}}+{\mathrm{MM}}_{\mathrm{Left}}+{\mathrm{TA}}_{\mathrm{Right}}+{\mathrm{TA}}_{\mathrm{Left}}}\times 100\;\left(\%\right)$$

The activity index may vary between +100% and −100%, where a negative number indicates a dominant TA activity and a positive number indicates a dominant MM activity.

Repeated measures analysis of variance (ANOVA) was performed for the comparison of serial periods. *p* < 0.05 was considered statistically significant. All data were analyzed using the Statistical Package for Social Sciences ver. 22.0 (SPSS, Chicago, IL, USA).

## 3. Results

### 3.1. T-Scan Analysis

No significant changes in OT or DT were noted during T0–T2 time intervals (Table 2). In addition, no significant anteroposterior occlusal force changes were noted, but there was a decrease in anterior dentition force and a concomitant increase in posterior dentition force (Figure 6). Through T0 to T2, the differences between the larger and smaller sides were 13%–20%, so there was some unbalance between left and right side; however, there were no significant changes from T0 to T2.

### 3.2. EMG Analysis

The changes between the voltage emitted by the MM and TA during time intervals were insignificant (Table 3). The mean value of EMG potentials of TA shows a higher value than MM. There were no marked predominance sides in the asymmetry index of both muscles, and the sign of the asymmetry index changed at each period. The activity index also showed negative values which mean a dominance of TA.

## 4. Discussion

Relapse and settling are examples of the craniofacial skeletal system adapting to a new occlusion. The achievement of a balance between morphological changes in occlusion and the functional adaptation of the surrounding neuromuscular system play a role in long-term occlusal stability [15]. Stable orthodontic retention depends on an equilibrium among the forces derived from the periodontal and gingival tissues, the orofacial musculature, the occlusion, and the post-treatment facial growth and development [20,21]. After achieving the desired occlusion through orthodontic treatment, several types of retainers are prescribed and applied to patients, including removable, fixed, passive, and active retainers. Unlike fixed retainers, removable retainers might affect the occlusal changes according to the kind of retainers.

There are several studies assessing occlusion after orthodontic treatment, but most studies have evaluated occlusal contacts with physical bite registration materials [2]. Recently, a few studies have used the T-scan system to evaluate the change of occlusal force. In 1987, the T-scan occlusal analysis system manufactured by Tekscan Inc. (South Boston, MA, USA) was developed by Professor William L. Maness. The T-scan system consists of a hand-held pressure-measuring sensor to be connected to a PC or laptop computer and software program that displays the occlusal changes on the screen. The pressure-measuring sensor is 100 μm of thickness. The T-scan system provides more reliable and improved information compared with traditional methods of occlusal assessment. In this system, not only occlusal contacts but also occlusal force and timing can be measured [22]. For functional occlusion evaluation after orthodontic treatment, Morton and Pancherz [23] examined patients’ mandibular movements with visual inspection clinically. Lustig et al. [24] studied occlusal contacts with wrap-around and clear-overlay retainers during a short period of retention with the T-scan system. They suggested that there were occlusal changes including decreased posterior force and occlusal surface area following placement of a full coverage VFR, while the wrap-around retainer showed increased posterior force and occlusal surface area. Desirable settling was inhibited by VFR. They concluded that the T-scan is a reliable method for occlusal analysis. Qadeer et al. [25,26] compared the occlusal force in post-orthodontic and non-orthodontic subjects with the T-scan system, but there was no consideration about retention protocols. They concluded that post-orthodontic subjects showed higher force percentages in the posterior area, and longer DT when compared to non-orthodontic subjects.

According to the T-scan system instructions, each OT and DT is ideally less than 0.2 s and 0.4 s because longer OT indicates more interference and premature contacts during closure and longer DT indicates more occlusal surface frictional contacts during excursion. The results of this study did not show the ideal OT and DT, but did show a range similar to other studies [25,26,27]. Lee and Lee [27] measured 48 healthy subjects by the T-scan system and averaged OT and DT. OT was similar with 0.2 s, but DTs were over 0.6 s.

In this study, there was a decrease in the anterior area force and a concomitant increase in the posterior. The increase in posterior force might be due to a vertical settling effect. This result might support our previous study, which showed the post-orthodontic settling in OTP with cast models [8]. Although all the incisal edges and occlusal surfaces were covered with removable retainers, the change of occlusion under retention period with OTP was different from that with VFR. VFR helps all the teeth including the second molars to stay at the final position at the end of orthodontic treatment. However, settling is inhibited with a VFR, and moreover, a posterior open bite might occur because of the inter-occlusal thickness of the retainer. To overcome the posterior open bite, Lindauer et al. modified the VFR to cover only anterior teeth and compared it to the Hawley retainer [28]. It caused extrusion of posterior teeth resulting in an anterior open bite. It was not considered to be successful settling. According to the results of this study, settling was allowed since the resin core was not extended to the second molars. The action of the mandibular tongue elevator retainer is also helpful. Seo et al. showed intrusion of posterior teeth with the tongue elevator in their 10-year follow-up study [29]. Despite the possible intrusion of mandibular molars, occlusal force of the posterior teeth increased, indicating a significant effect of settling of the maxillary molars. Lustig et al. [24] reported the force decrease in the anterior area might be due to a physiologic rebound of stresses created from forces of finishing elastics or an effect seen following posterior settling. In any case, heavy anterior forces are contradictory to good gnathologic function.

There are two reasons that the second molars matter in settling procedures. First, we prefer to finish the orthodontic treatment with Tweed occlusion or transitional occlusion. This occlusion is characterized by functional disclusion of the second molars. In the retention period, the second molars can re-erupt to a healthy functional occlusion without trauma or premature contact with balanced action of masticatory muscles [30]. Second, the inter-occlusal thickness of full-coverage retainers might cause increased contact of the terminal teeth since they are at the back end of a functional wedge. Then the result is second molar infra-occlusion in the retention period. For these reasons, one of the most important factors of the OTP is that the resin core does not extend to the second molars, allowing the occlusal settling.

Meanwhile, according to the contraction and relaxation of muscles, the electrical potential changes. EMG uses these quantitative data. EMG has been used to diagnose and plan the treatment for temporomandibular disorders (TMD), and is now considered as a useful tool for diagnosis and determination of treatment outcomes [31]. There are two types of EMG: intramuscular EMG and surface EMG. Intramuscular EMG uses needles and fine-wire electrodes. They are inserted through the skin into the muscle tissue, and detect single motor unit action potential. In contrast, surface EMG (sEMG) uses surface electrodes and they are attached on the skin with a patch. They detect superimposed motor unit action potential from many fibers. The masseter (MM) and temporalis anterior (TA) are located close to the skin; these muscles are easy to access with sEMG. Static activity, such as rest, or clenching and dynamic activity such as opening/closing the mouth, protrusion, left/right excursion, chewing, mastication, swallowing, or speaking can be recorded and assessed [32,33]. Ferrario et al. mentioned that sEMG of masticatory muscles may provide the orthodontists with a fast, low-cost, and non-invasive quantitative test to predict post-orthodontic stability [34]. Kerstein and Radke [35] also proposed that these systems can help with occlusal adjustment and, if it is properly performed, significant muscle activity level reductions and reduction of disclusion time may be immediately observed.

EMG analysis in this study showed that the potentials of TA were higher than that of MM in all periods. This is consistent with the other studies which investigated the EMG [36,37,38]. Wieczorek et al. [36] found that the voltage recorded in TA is higher than that in MM in healthy subjects. Ferrario et al. [37] showed that TA potentials were greater than those of MM in a centric occlusion state with healthy normal young people, but not in a clench test.

In this study, there was no marked predominance in the asymmetry index of both muscles. This might imply that the imbalance of occlusal force does not correlate with the distribution of muscle activity. EMG studies [19,39,40,41] evaluated healthy and asymptomatic subjects who showed asymmetric muscle activity and concluded that unbalanced or asymmetrical activity is not a muscle pathology in healthy subjects. Naeije et al. [19] showed that the asymmetry in muscle activity depends on the clenching level. Scopel et al. [40] found that even normal subjects with sound dentition have a physiologically asymmetrical muscle activity, and an asymmetry and activity index up to 4%–17% may be considered as normal function without symptoms.

In young and healthy people, mandibular posture is usually more controlled by TA than MM [37]. In this study, the mean of activity index also showed a dominance of TA: 13 and 15 of a total 18 subjects, at each T0 and T1, respectively, showed a negative value that means a dominance of TA.

The value of this study is that advanced measuring methods were used to evaluate the change after orthodontic treatment. Almost all studies with the T-scan system have been limited to the field of prosthetic dentistry, where, using the computer-displayed occlusal data, proper occlusal adjustments are made and interferences can be reduced in the natural dentition, dental prostheses, or dental implants [42]. EMG has been used in other fields than orthodontics. These systems have been grafted in the field of orthodontics for several authors, and we confirmed that the occlusal and muscular analysis in the retention period can be achieved in an easy and intuitive way. At every follow-up visit, data can be acquired easily with a T-scan system and EMG. In summary, analyzing simultaneous occlusal forces and muscle activity with the T-scan system and EMG system demonstrates synergic effects and gives a clear understanding of the patients’ occlusal conditions. Even though this was a short-term evaluation of occlusion and muscle activity, significant changes were not shown in occlusion or muscle activity.

This preliminary study was based on the quantitative change of occlusion in four-month retention periods with 18 subjects. As a pilot study to evaluate the possibility of OTP, methodological limitations were inevitable. Long-term studies with more subjects in subgroups are required to evaluate the long-term effects of the OTP and tongue elevator on the occlusal and muscular changes, based on initial characteristics of samples such as age, gender, skeletal pattern, and malocclusion type. In addition, the establishment of a control group is needed in a further study. The method we used in this study will be helpful for our further studies and for any other study of occlusion and muscular activity. The OTP, despite its design to cover the occlusal surface, did not cause any unfavorable changes in occlusion, which is a typical problem of clear retainers. The tendency of the posterior occlusal force to increase and the anterior occlusal force to decrease relatively is considered to be desirable settling. There were no significant changes of EMG activity during a four-month use of OTPs. The OTP can be a good alternative for clear retainers in the orthodontic field.

## 5. Conclusions

Based on the findings from the preliminary study, the null hypothesis was accepted because the occlusion time and disclusion time did not significantly change during four-month orthodontic retention periods, and varied in normal ranges. Occlusal forces decreased in the anterior dentition and concomitantly increased in the posterior dentition during the four months after retainer delivery. There was no clear evidence of a relationship between unbalanced occlusal forces and muscle activity. In most subjects, the temporalis anterior muscle was more dominant than the masseter muscle.

## Figures and Tables

**Figure 1 sensors-21-00541-f001:**
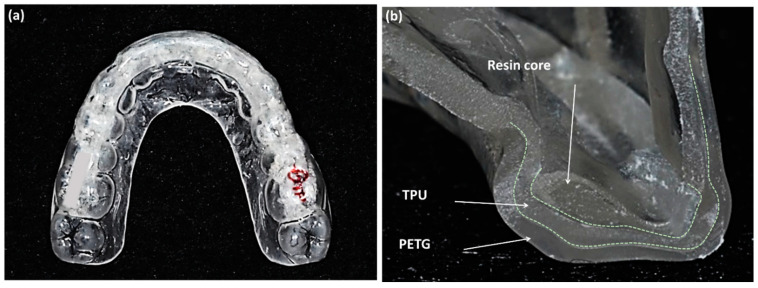
(**a**) Oral-treaper (OTP) retainer for maxillary dentition; (**b**) the three layers of the OTP on the incisor at the sliced surface, reinforced resin core, thermoplastic polyurethane (TPU), and polyethylene terephthalate glycol (PETG) (Forestacryl and Track B thermoforming foils, Forestadent, Bernard Förster GmbH, Pforzheim, Germany).

**Figure 2 sensors-21-00541-f002:**
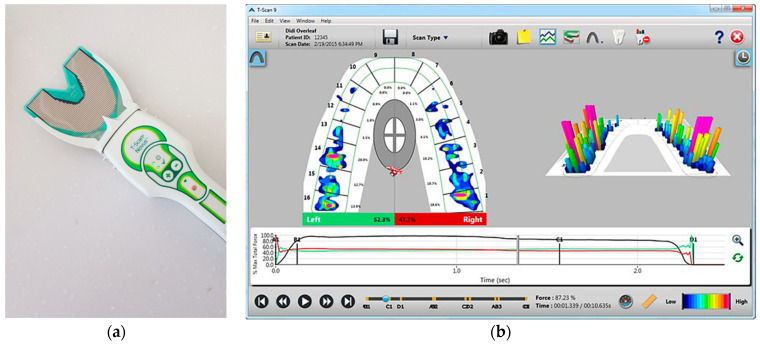
T-Scan system (T-Scan Novus system). (**a**) Scanner with a hand-held pressure-measuring sensor to be connected to a PC or notebook; (**b**) T-scan v10 software displaying the occlusal changes. As soon as the scan is completed, it is displayed on the screen, and the scan is positioned at the point of initial contact. The 2D force-view shows the distribution of occlusal contacts. While the green box shows a proportion of occlusal forces on the left side, the red box shows that on the right side. Occlusal forces and changes are displayed with a colored map according to the reference color scale on the lower right corner. The graph below shows the change of occlusal distribution based on time.

**Figure 3 sensors-21-00541-f003:**
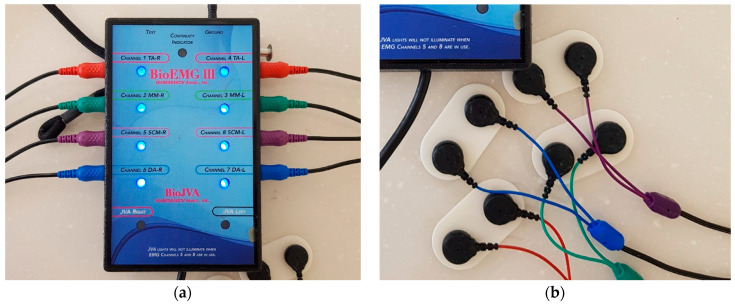
Electromyography (EMG) system (BioEMG system). (**a**) A device that collects the electrical signals and measures and amplifies the electric potential. The red socket is connected to the electrodes attached to the temporalis anterior muscle, the green one is connected to the electrodes attached to the masseter muscle, and the purple one and the blue one collect and measure the electric potential from the sternocleidomastoid and digastric muscles, respectively. (**b**) The electrodes with the sensors are attached with a patch to the skin near each muscle. (**c**) A computer screen provides the information of muscle activity. T-scan data (left) and EMG data (right) can be collected concurrently.

**Figure 4 sensors-21-00541-f004:**
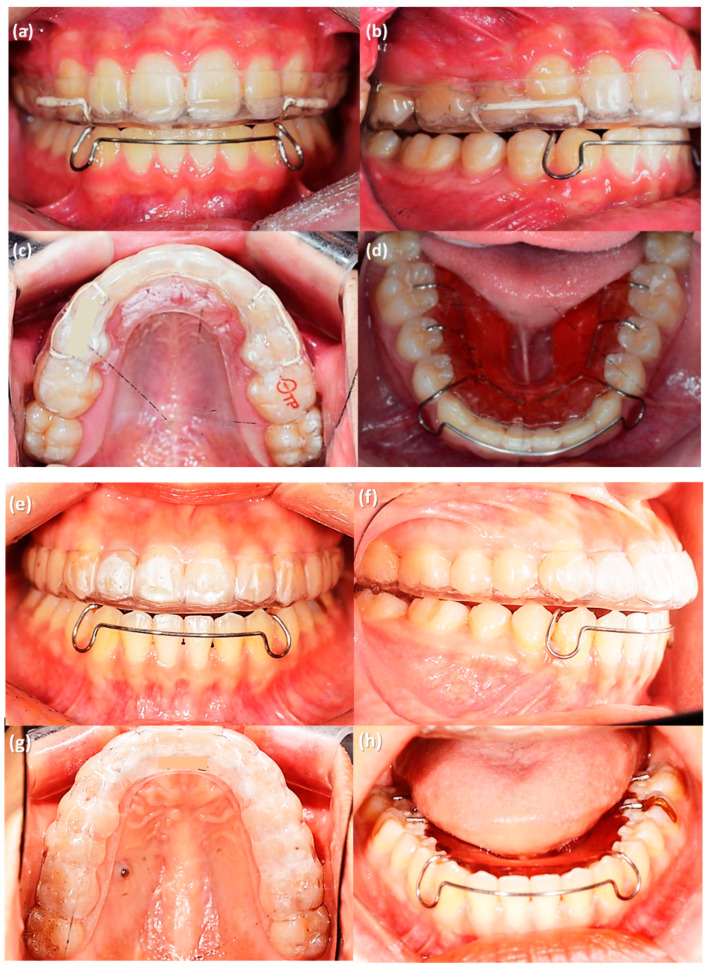
Removable retainers provided to the patients after orthodontic treatment. (**a**–**d**) Premolar extraction case, OTP retainer in the maxillary dentition (**a**–**c**) and tongue elevator for mandibular dentition (**d**). (**e**–**h**) Non-extraction case, OTP retainer in the maxillary dentition (**e**–**g**) and tongue elevator for mandibular dentition (**h**).

**Figure 5 sensors-21-00541-f005:**
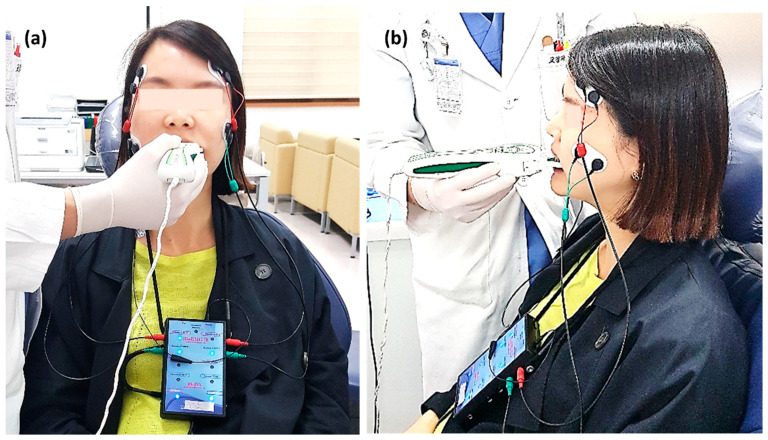
(**a**) T-scan system (Tekscan Inc., South Boston, MA, USA) and electromyography (EMG) recording device (BioEMG III^TM^, BioResearch Inc., Milwaukee, WI, USA) application to a patient. The patient sits in an upright position and bilateral electrodes are placed on the temporalis anterior (TA) and masseter (MM) muscles, simultaneously with the biting T-scan sensor. (**b**) The position of electrodes attached to the TA and MM muscles is shown on the side view. (**c**) Computer display of the occlusal force and muscle activity at maximal intercuspal position (MICP).

**Figure 6 sensors-21-00541-f006:**
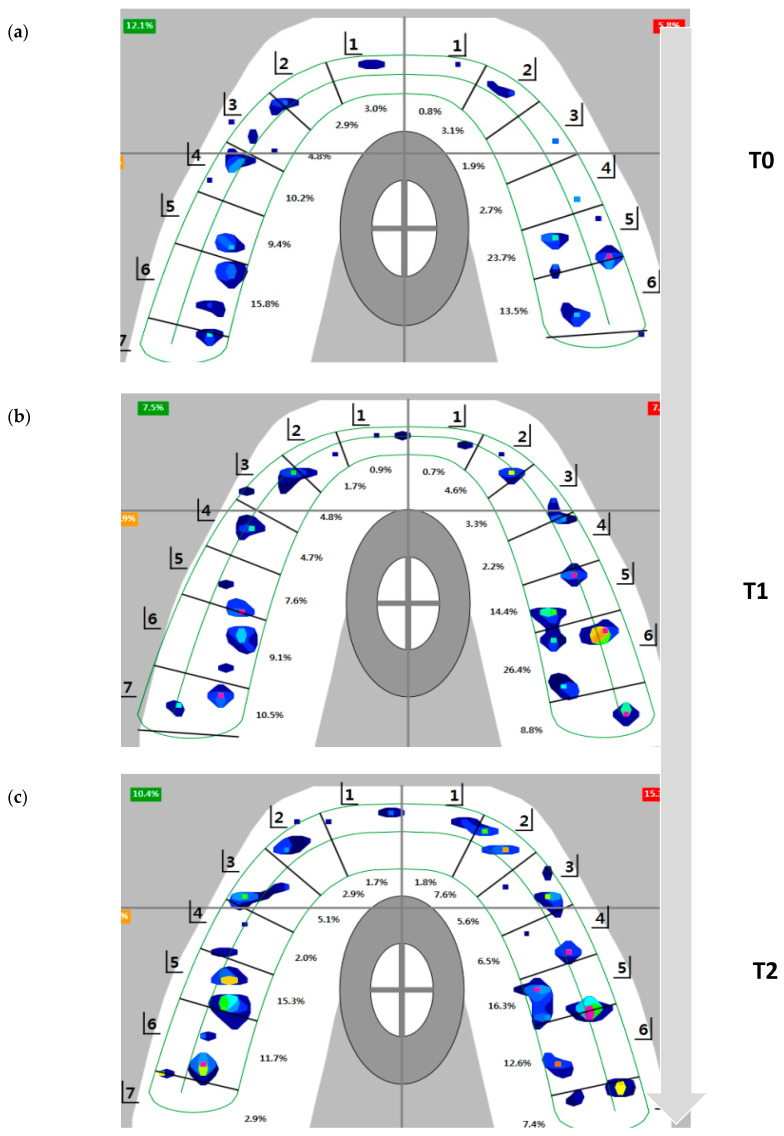
Flow of occlusal contact view during T0 to T2 in T-scan system. The data were analyzed by calculating the total force per bite registration and determining the percentage of force by quadrant (total 100%) for each centric occlusion bite at MICP. (**a**) T0, at debonding. (**b**) T1, one month after retainer delivery. (**c**) T2, four months after retainer delivery.

**Table 1 sensors-21-00541-t001:** Demographic data of subjects.

	Subjects
Number (n)	18
Age (years)	20.3 ± 6.0 ^1^
Gender (n)	
Male	8
Female	10
Sagittal pattern (n)	
Class I	8
Class II	6
Class III	4
Vertical pattern (n)	
Hyperdivergent	3
Normodivergent	10
Hypodivergent	5
Treatment option (n)	
Premolars extraction	6
Non-extraction	12

^1^ Mean ± SD (standard deviation).

**Table 2 sensors-21-00541-t002:** Occlusal time and percentage of force distribution in subjects from T0 and T2.

	T0		T1		T2		*p*-Value ^5^
	Mean	SD	Mean	SD	Mean	SD	
Time (s)							
OT ^1^	0.23	0.12	0.27	0.13	0.24	0.15	0.494
Pro-DT ^2^	0.52	0.29	0.44	0.17	0.56	0.27	0.279
Left-DT ^3^	0.95	0.49	0.72	0.29	0.81	0.43	0.070
Right-DT ^4^	0.88	0.61	0.75	0.60	0.58	0.25	0.135
Force (%)							
Left Anterior	7.68	8.04	6.01	6.60	4.88	5.30	0.226
Right Anterior	7.57	7.56	5.77	4.69	5.07	5.48	0.269
Anterior	15.24	14.73	11.78	10.41	9.95	9.78	0.213
Left Posterior	49.68	9.60	47.08	10.03	49.66	12.55	0.488
Right Posterior	35.07	10.36	41.13	9.81	40.38	9.49	0.010 * T1 > T0 (0.020), T2 > T0 (0.042) ^6^
Posterior	84.76	14.73	88.21	10.40	90.04	9.76	0.215
Larger side	57.47	6.83	56.86	6.01	59.09	6.30	0.234
Smaller side	42.53	6.85	43.16	6.02	40.90	6.31	0.233

^1^ Occlusion time at MICP; ^2^ disclusion time at protrusive movement; ^3^ disclusion time at left excursive movement; ^4^ disclusion time at right excursive movement; ^5^ repeated measures ANOVA test; ^6^ the Bonferroni correction sets the significance cut-off at 0.05; * *p* < 0.05.

**Table 3 sensors-21-00541-t003:** Change of EMG potentials in subjects from T0 and T2.

	T0		T1		T2		*p*-Value ^8^
	Mean	SD	Mean	SD	Mean	SD	
Potentials (μV)							
TA_L ^1^	79.00	39.31	71.33	27.32	90.89	42.91	0.202
TA_R ^2^	85.59	44.53	88.56	37.68	87.61	41.71	0.934
MM_L ^3^	74.53	61.08	67.17	65.74	72.44	47.94	0.613
MM_R ^4^	61.24	51.25	62.00	55.35	56.94	29.47	0.983
Asymmetry index (%)							
Asym_MM ^5^	4.83	28.03	−1.73	32.73	2.79	25.75	0.891
Asym_TA ^6^	−3.04	20.26	−2.17	22.04	12.06	21.37	0.089
Asym index ^7^	0.31	19.32	−0.93	21.38	9.84	14.92	0.248
Activity index	−18.21	29.07	−19.92	27.76	−17.59	27.25	0.939

^1^ Temporalis EMG activity on the left side; ^2^ temporalis EMG activity on the right side; ^3^ masseter EMG activity on the left side; ^4^ masseter EMG activity on the right side; ^5^ asymmetry index of masseter; ^6^ asymmetry index of temporalis; ^7^ asymmetry index of both masseter and temporalis; ^8^ repeated measures ANOVA test.

## Data Availability

Not applicable.

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
