# Peer review of "Computerized Assessment of Occlusion and Muscle Activity during Use of a Multilayer Clear Retainer: A Preliminary Study"

_sensors, 2021, doi:10.3390/s21020541_

Round 1

Reviewer 1 Report

Dear authors,

We have read with interest your clinical study occlusal contact changes after orthodontic treatments.

The methods used for investigation are precise and can bring valuable informations.

However, we clearly miss the deep or consistent objective of your clinical investigation:

Were you trying to investigate the innoccuity of the OTP device? Or were you really willing to demonstrate OTP efficiency (to prevent occlusal contact changes)?

The Title of your manuscript describe a "preliminary study". But we can't get the meaning of your experiments. Which parameters would you investigate to go further, to continue this "preliminary study"?

You get some precious quantitative data, but without possibility to get some conclusion.

Our recommandation is that you define the objective of your study, in accordance with the methods used. And in order to use (and discuss) reliable informations, toward further complete clinical investigation.

At the end of the 'Introduction' section, you write: "The aim of the present study was to evaluate the changes in occlusion and masticatory muscle activity during the use of a specific type of retainer after orthodontic treatment."

But without control group, this aim is unreachable (how could you know that your observations are due to the retainer device?). Moreover, the very limited number of subjects (especially if we take in account the various profiles - Type I / II / III) prevent any reliable conclusions.

We still consider that the overall work is worth sharing with the scientific / medical community, after deep re-analysis and discussion.

At that stage, the obtained conclusion are too poor to bring any info the the community: "In conclusion, during a retention period with the multi-layer clear retainer, there were no significant changes in the occlusion forces, asymmetry or activity indexes of muscle activity."

Author Response

Dear Editor,

We would like to thank the reviewers for their time spent on reviewing our manuscript and their valuable comments helping us improving the article.

In this document, we have responded to the issues raised by the editor in a point-by-point fashion. Comments from the Associate editor and reviewers are in black color while our response is in red color. We sincerely hope that we have addressed all of your concerns.

  • Reviewer 1:

Dear authors,

We have read with interest your clinical study occlusal contact changes after orthodontic treatments.

The methods used for investigation are precise and can bring valuable information. However, we clearly miss the deep or consistent objective of your clinical investigation:

Were you trying to investigate the innoccuity of the OTP device? Or were you really willing to demonstrate OTP efficiency (to prevent occlusal contact changes)?

  • We appreciate Reviewer #1’s considerate comments.
  • The purpose of this study was to investigate the innocuousness of OTP retainer, rather than to demonstrate the efficiency of OTP retainer. We clarified the aim of this study in the last paragraph of the introduction part.

The Title of your manuscript describe a "preliminary study". But we can't get the meaning of your experiments. Which parameters would you investigate to go further, to continue this "preliminary study"?

  • We appreciate Reviewer #1’s considerate comments.
  • This pilot study showed the possibility that OTPs can be a good removable orthodontic retainer for long-term use, even though it has full coverage of the occlusal surface. Using the same parameters, we will try to evaluate more long term data and any differences based on initial characteristics of samples (such as age, gender, skeletal pattern, and malocclusion type) in further study.
  • We referred in the last paragraph of the discussion part.

You get some precious quantitative data, but without possibility to get some conclusion. Our recommandation is that you define the objective of your study, in accordance with the methods used. And in order to use (and discuss) reliable informations, toward further complete clinical investigation.

  • We appreciate Reviewer #1’s considerate comments.
  • We revised the objective or this study, considering your overall comments.

At the end of the 'Introduction' section, you write: "The aim of the present study was to evaluate the changes in occlusion and masticatory muscle activity during the use of a specific type of retainer after orthodontic treatment." But without control group, this aim is unreachable (how could you know that your observations are due to the retainer device?). Moreover, the very limited number of subjects (especially if we take in account the various profiles - Type I / II / III) prevent any reliable conclusions.

  • We appreciate Reviewer #1’s considerate comments.
  • We strongly agree with Reviewer #1’s comments. This was the first clinical trial for long-term use of OTPs to evaluate functional outcomes such as occlusal changes and muscle activity. As a pilot study to evaluate the possibility of OTP, methodological limitations inevitably existed. The establishment of control group and securing a sufficient number of samples in subgroup will be conducted in a further study.
  • We referred it in the last paragraph of the discussion part as well.

We still consider that the overall work is worth sharing with the scientific / medical community, after deep re-analysis and discussion.

At that stage, the obtained conclusion is too poor to bring any info the community: "In conclusion, during a retention period with the multi-layer clear retainer, there were no significant changes in the occlusion forces, asymmetry or activity indexes of muscle activity."

  • We appreciate Reviewer #1’s considerate comments.
  • The conclusion was rewritten as follows;

The OTP, despite its design to cover the occlusal surface, did not cause any unfavorable changes in occlusion, which is typical problem of clear retainers. The tendency of the posterior occlusal force to increase and the anterior occlusal force to decrease relatively is considered to be desirable settling. There were no significant changes of EMG activity for 4 month-use of OTPs. The OTP can be a good alternative of clear retainers in orthodontic field.

Reviewer 2 Report

This work is interesting on a specific topic, however, suggest some changes and corrections

  • The introduction section is too long and first page is not focused on the topic of the work. It should be reconstructed and shortened.
  • Has the sample size necessary to conduct the research been calculated? If not, I suggest that you calculate the power of the tests so that readers are aware of how representative the reported relationships are.
  • The data shown in Fig. 7 is the same as in Table 2, and even it is a lean version thereof. I suggest either to delete Fig 7 as it adds nothing, or prepare it more carefully (showing all the necessary data and results) and delete the table 2. The presented results should not be duplicated in different forms.

Author Response

Dear Editor,

We would like to thank the reviewers for their time spent on reviewing our manuscript and their valuable comments helping us improving the article.

In this document, we have responded to the issues raised by the editor in a point-by-point fashion. Comments from the Associate editor and reviewers are in black color while our response is in red color. We sincerely hope that we have addressed all of your concerns.

  • Reviewer 2:

This work is interesting on a specific topic, however, suggest some changes and corrections

The introduction section is too long and first page is not focused on the topic of the work. It should be reconstructed and shortened.

  • We appreciate Reviewer #2’s considerate comments.
  • We reorganized paragraphs and shortened the length in Introduction. Reorganized paragraphs of the introduction section is following:
    • 1st paragraph: the meaning and importance of retention and settling after orthodontic treatment
    • 2nd paragraph: introduction of various removable retainers, and pros and cons of vacuum formed retainer (VFR)
    • 3rd paragraph: the characteristics of Oral-treaper (OTP) and its expected advantages for orthodontic retention
    • 4th paragraph: evaluation of occlusion and muscle activity by T-scan and electromyography (EMG)
    • 5th paragraph: the aim and hypothesis of this study on the final paragraph.
  • Detailed explanation about T-scan and EMG was moved to the discussion part.

Has the sample size necessary to conduct the research been calculated? If not, I suggest that you calculate the power of the tests so that readers are aware of how representative the reported relationships are.

  • We appreciate Reviewer #2’s considerate comments.
  • We conducted the calculation of the power for evaluating the sample size. It has been added to the subjects’ part in the materials and methods section.

The data shown in Fig. 7 is the same as in Table 2, and even it is a lean version thereof. I suggest either to delete Fig 7 as it adds nothing, or prepare it more carefully (showing all the necessary data and results) and delete the table 2. The presented results should not be duplicated in different forms.

  • We appreciate Reviewer #2’s considerate comments.
  • We removed Figure 7 (integrated into Table 2) and Figure 8 (integrated into Table 3).

Reviewer 3 Report

Dear authors, I read your article carefully. 

I have certain questions about the methodology.

 You wrote that the inner layer does not extend to second molar to allow that tooth to be settled. You cited your own article. I wonder why you think the  second molar is  came to infraocclusion? 

We also feel that groups are too small if they are divided by classes. 

LINE 254 AGE 20.3 ± 6.0 SD or SE? - LINE 322 EQUALIZE 0.05 OR .05 - LINE 385 TWICE WRITES MASSETER EMG ACTIVITY ON THE RIGHT

Author Response

Dear Editor,

We would like to thank the reviewers for their time spent on reviewing our manuscript and their valuable comments helping us improving the article.

In this document, we have responded to the issues raised by the editor in a point-by-point fashion. Comments from the Associate editor and reviewers are in black color while our response is in red color. We sincerely hope that we have addressed all of your concerns.

  • Reviewer 3:

Dear authors, I read your article carefully.

I have certain questions about the methodology.

 You wrote that the inner layer does not extend to second molar to allow that tooth to be settled. You cited your own article. I wonder why you think the second molar is came to infraocclusion?

  • We appreciate Reviewer #3’s considerate comments.
  • There are two reasons that the second molars matter in settling procedures. First, we prefer to finish the orthodontic treatment with Tweed occlusion or transitional occlusion. This occlusion is characterized by disclusion of the second molars. In the retention period, the second molars can re-erupt to a healthy functional occlusion without trauma or premature contact with balanced action of masticatory muscles [33]. Second, the inter-occlusal thickness of full-coverage retainers might result in the increased contact of the terminal teeth by “wedge effect” concept. Then the second molars are prone to be affected with infra-occlusion in the retention period. For these reasons, one of the most important factors of OTP is that the resin core does not extend to the second molars, allowing the occlusal settling.
  • We added this explanation in the discussion part.

We also feel that groups are too small if they are divided by classes.

  • We appreciate Reviewer #3’s considerate comments.
  • We agree to Reviewer #3 comments that sagittal relationship can influence the distribution of occlusal force and muscle activities. In this study, we showed the demographic data of sagittal pattern in Table 1. However, the sample size is too small to compare between subgroup divided by sagittal pattern, all statistical analysis was performed for the entire samples. We plan to secure more samples and compare between subgroups based on the initial skeletal pattern and malocclusion type in future study.

LINE 254 AGE 20.3 ± 6.0 SD or SE? - LINE 322 EQUALIZE 0.05 OR .05 - LINE 385 TWICE WRITES MASSETER EMG ACTIVITY ON THE RIGHT

  • We appreciate Reviewer #3’s considerate comments.
  • LINE 254: The age data was described as mean ± SD (standard deviation). We added this information as a footnote.
  • LINE 322: The P-value was unified as05.
  • LINE 385: We corrected the footnote.

Round 2

Reviewer 1 Report

Dear Authors,

As already expressed after the reading of the original manuscript, we appreciate the clinical investigation you have conducted in this project.

Your answers to our questions/remarks are quite clear and the modifications/addition to the revised manuscript improve the overall readability (to our point of view).

Therefore, we recommend the editor to accept your revised manuscript

Author Response

Dear Editor,

We would like to thank the reviewers for their time spent on reviewing our manuscript and their valuable comments helping us improving the article.

In this document, we have responded to the issues raised by the editor in a point-by-point fashion. Comments from the Associate editor and reviewers are in black color while our response is in red color. We sincerely hope that we have addressed all of your concerns.

  • Reviewer 1:
  • Thank you for revising the work. There are a few minor comments remaining: Title is too long, and somewhat confusing. How about shortening it to: Computerized assessment of occlusion and muscle activity during use of a multilayer clear retainer: A preliminary study
  • We appreciate Reviewer #1’s considerate comments. We changed the title as the reviewer suggested. “Computerized assessment of occlusion and muscle activity during use of a multilayer clear retainer: A preliminary study”
  • Main conclusions need some rewriting: "In this preliminary study, within a 4-month orthodontic retention period, the occlusion time and disclusion time did not significantly during retention periods, and varied in normal ranges. Occlusal forces decreased in the anterior dentition and concomitantly increased in the posterior dentition during the 4 months after retainer delivery. There was no clear evidence of a relationship between unbalanced occlusal forces and muscle activity. In most subjects, the temporalis anterior muscle was more dominant than the masseter."
  • Please move the next sentence “The OTP, despite its design to cover the occlusal surface … “ back to the summary paragraph in the end of discussion section since this conclusion is too broad for the limited study performed.
  • We appreciate Reviewer #1’s considerate comments. We rewrote the conclusion and moved the sentence to the end of discussion section as the reviewer commented.